# Transcriptomic and Metabolomic Mechanisms Underlying Adaptive Differentiation of Black Soldier Fly Larvae Induced by Regional Food Waste Domestication

**DOI:** 10.3390/biology14111584

**Published:** 2025-11-12

**Authors:** Bin Zhang, Rencan Yang, Zaimei Yuan, Hongren Su, Jingyi Shi, Shichun He, Sifan Dai, Dawei Sun, Zhiyong Zhao, Qingquan Hu, Dongwang Wu

**Affiliations:** 1Yunnan Academy of Animal Husbandry and Veterinary Sciences, Kunming 650051, China; 2Kunming Animal Disease Prevention and Control Center, Kunming 650106, China; 3Yunnan Provincial Key Laboratory of Animal Nutrition and Feed, Faculty of Animal Science and Technology, Yunnan Agricultural University, Kunming 650201, China

**Keywords:** *Hermetia illucens*, regional food waste domestication, molecular adaptation, transcriptomics, metabolomics

## Abstract

Black soldier flies are efficient at decomposing organic waste, and their adaptability to different food sources is key for practical use. This study investigated the molecular adaptations of black soldier fly larvae after long-term domestication on regional food waste. It was found that the larvae developed distinct gene expression and metabolite profiles to adapt to specific food waste characteristics. Key adaptations included adjustments in detoxification pathways, nutrient metabolism, and stress response mechanisms, with certain transcription factors and unannotated metabolites playing important roles. These findings reveal how black soldier flies adapt to regional food waste, providing a basis for optimizing their use in waste treatment and strain improvement.

## 1. Introduction

With the acceleration of urbanization, the production of kitchen waste (food waste) has been increasing year by year. Its characteristics of high water content, high organic matter, and complex composition not only exert significant environmental pressure but also lead to resource waste [1,2]. Traditional disposal methods such as landfilling and incineration are prone to causing secondary pollution. In contrast, insect conversion technology, as a novel biological treatment method [3], has become a research hotspot in recent years due to its advantages of high efficiency in waste reduction and resource recovery. Among various insects, the larvae of the black soldier fly (*Hermetia illucens*) have been widely used for the biodegradation of kitchen waste owing to their broad diet, rapid growth, and high conversion efficiency. Insect biomass can serve as a high-quality protein feed [4,5], while frass (insect excrement) can be utilized as organic fertilizer, achieving a circular utilization model of “waste-insect-resource” [6].

The composition of kitchen waste is significantly influenced by regional dietary habits and types of food establishments, leading to notable differences in the nutritional structure (e.g., the ratio of carbohydrates, fats, and proteins) of food waste across different regions [7]. Variations in the substrate may directly affect the growth, development, metabolic adaptation, and nutrient accumulation efficiency of black soldier fly larvae [8,9,10]. On the one hand, changes in substrate composition can alter the larvae’s feeding preferences and digestive efficiency, thereby influencing growth performance indicators such as weight gain and developmental duration. On the other hand, in response to potential specific harmful substances in food waste, the larvae’s detoxification metabolic systems (such as pathways involving cytochrome P450 and glutathione S-transferase) may activate adaptive regulatory mechanisms. Meanwhile, differences in substrate nutrition could also lead to variations in the accumulation patterns of nutrients such as proteins, fatty acids, and amino acids in the larvae, ultimately affecting their resource utilization value [11,12].

Currently, research on the utilization of black soldier fly larvae for the treatment of kitchen waste primarily focuses on optimizing processing efficiency and evaluating the nutritional value of the larvae. However, studies investigating the specific effects of food waste from different regional sources on the larvae—particularly those elucidating the differences in growth performance, responses of detoxification metabolic pathways, and regulatory mechanisms of nutrient accumulation at the molecular level—remain relatively limited [13]. Furthermore, it is essential to explore whether black soldier fly populations that have long-term adapted to region-specific food waste have undergone adaptive evolution in their metabolic pathways, and how such adaptations may influence their efficiency in processing homologous or heterologous food waste. These scientific questions urgently require in-depth investigation.

This study selected populations of black soldier fly larvae that have been long-term processing food waste from the urban area of Kunming City and the Qilin District of Qujing City (Group A and Group C) as research subjects. By controlling breeding density and processing procedures, the study systematically compared the differences in growth performance between the two groups of larvae in their respective regional food waste environments. Molecular biology techniques were employed to investigate the regulatory mechanisms of their detoxification metabolic pathways, and the characteristics of nutrient accumulation in the larvae were analyzed. The aim is to reveal the molecular regulatory mechanisms of food waste from different regions on black soldier fly larvae, provide a theoretical basis for optimizing the processing technology of black soldier flies for region-specific kitchen waste and enhancing the resource utilization value of the larvae, while also offering new perspectives for understanding the adaptation mechanisms of insects to heterogeneous organic waste.

## 2. Materials and Methods

### 2.1. Black Soldier Fly Breeding Experiment

The black soldier fly (*Hermetia illucens*) larvae used in this experiment were derived from a single base population maintained by the Yunnan Academy of Animal Science and Veterinary Medicine (Panlong District, Kunming, China). This population has been adapted to feed on restaurant food waste. Since March 2022, the base population was divided into two groups, A and C, and reared continuously under different geographical conditions. Group A was consistently reared at the original site of the Yunnan Academy of Animal Science and Veterinary Medicine and fed with food waste from restaurants in Panlong District, Kunming. Group C was transferred to a laboratory in Qilin District, Qujing, and fed with local food waste. Experiments were conducted simultaneously in laboratories in Kunming and Qujing. Both Group A and Group C were reared in identical 4 m^2^ experimental setups, with three replicates per group. Group A was fed a total of 135 kg of food waste collected from large restaurants in Panlong District, Kunming, while Group C was fed a total of 135 kg of food waste from households and small to medium-sized restaurants in Qilin District, Qujing (Table 1 and Table 2). All food waste was pre-treated as follows: first subjected to three-phase separation, then the remaining solid residue was ground into a slurry, and the moisture content was adjusted to 77–85%.

Prior to the experiment, newly hatched larvae from both Group A and Group C were fed a larval diet (consisting of soybean meal, corn, and wheat bran mixed in a 1:2:3 ratio, with moisture content adjusted to approximately 75% using water) until they reached 5 days of age. The formal experiment commenced on 19 October 2022, when healthy 5-day-old larvae were introduced at a density of 250,000 individuals per square meter into the experimental devices containing a quantified amount of pre-treated food waste. During the feeding process, additional food waste was supplemented as needed based on the larvae’s degradation progress, until the waste was completely converted into sand-like frass.

On day 3 of the experiment (i.e., when the larvae were 8 days old, 21 October 2022), sample collection and preparation were carried out. First, 50 larvae were randomly selected from each replicate and weighed to assess their growth status (Figure 1). Subsequently, an additional 60 larvae were randomly collected from each group for molecular biological analysis. The collected larvae were instantaneously euthanized with chloroform, rinsed five times with sterile water, and subjected to tissue separation under aseptic conditions. The tissues of every 10 larvae were pooled into one biological replicate sample, aliquoted into 2 mL cryotubes, rapidly frozen in liquid nitrogen, and stored in a −80 °C ultra-low temperature freezer for subsequent RNA extraction and metabolomic analysis. The two types of food waste are shown in Table 1 and Table 2.

### 2.2. Sample Extraction and Analysis

Total RNA was extracted from black soldier fly larval tissues using TRIzol^®^ reagent (Magen, Guangzhou, China). The A260/A280 absorbance ratio of the RNA samples was measured using a Nanodrop ND-2000 (Thermo Scientific, Waltham, MA, USA), and the RNA integrity number (RIN) was determined using an Agilent Bioanalyzer 4150 (Agilent Technologies, Santa Clara, CA, USA). Only RNA samples that passed quality control were used for library construction. A paired-end (PE) library was prepared according to the instructions of the ABclonal mRNA-seq Lib Prep Kit (ABclonal, Wuhan, China). mRNA was purified from 1 μg of total RNA using oligo(dT) magnetic beads and then fragmented in ABclonal First Strand Synthesis Reaction Buffer. The fragmented mRNA was used as a template to synthesize first-strand cDNA with random primers and reverse transcriptase (RNase H), followed by second-strand cDNA synthesis using DNA polymerase I, RNase H, buffer, and dNTPs. The synthesized double-stranded cDNA fragments were ligated with adapter sequences for PCR amplification. The PCR products were purified, and library quality was assessed using an Agilent Bioanalyzer 4150 (Agilent Technologies, Santa Clara, CA, USA). Finally, sequencing was performed on an Illumina NovaSeq 6000 sequencing platform (Illumina, San Diego, CA, USA).

Metabolites Extraction: 100 μL of sample was transferred to an EP tube. After the addition of 400 μL of extract solution (acetonitrile–methanol = 1:1, containing isotopically-labelled internal standard mixture), the samples were vortexed for 30 s, sonicated for 5 min in an ice-water bath, and incubated for 1 h at −40 °C to precipitate proteins. Then the sample was centrifuged at 12,000 rpm for 15 min at 4 °C. The resulting supernatant was transferred to a fresh glass vial for analysis. The quality control (QC) sample was prepared by mixing an equal aliquot of the supernatants from all of the samples.

### 2.3. Library Construction and Sequencing

A paired-end (PE) sequencing library was prepared following the manufacturer’s instructions of the ABclonal mRNA-seq Lib Prep Kit (ABclonal, Wuhan, China). Briefly, mRNA was purified from 1 μg of total RNA using oligo(dT) magnetic beads and subsequently fragmented in ABclonal First Strand Synthesis Reaction Buffer. The fragmented mRNA was then used as a template to synthesize first-strand cDNA with random primers and reverse transcriptase (RNase H), followed by second-strand cDNA synthesis using DNA polymerase I, RNase H, buffer, and dNTPs. The resulting double-stranded cDNA fragments were ligated with adapter sequences and amplified by PCR. The PCR products were purified, and library quality was assessed using an Agilent Bioanalyzer 4150 (Agilent Technologies, Santa Clara, CA, USA). Finally, sequencing was performed on an Illumina NovaSeq 6000 platform (Illumina, San Diego, CA, USA), and the generated data were used for bioinformatic analysis.

### 2.4. Data Analysis

Raw sequencing data generated from the Illumina platform were initially processed to ensure quality. Adapter sequences and low-quality reads were removed using fastp (v0.23.2) with default parameters. The clean reads were then aligned to the reference genome of *Hermetia illucens* (NCBI Assembly: ASM2896936v1) using HISAT2 (v2.2.1). Gene expression levels were quantified as FPKM (Fragments Per Kilobase of transcript per Million mapped reads) using StringTie (v2.2.1). Differential gene expression analysis was performed with DESeq2 (v1.38.3), applying a threshold of |log2FoldChange| ≥ 1 and adjusted *p*-value < 0.05 to identify significantly differentially expressed genes. For metabolomic data, raw mass spectrometry files were processed with MS-DIAL (v4.9) for peak picking, alignment, and metabolite identification based on the HMDB and KEGG databases. Significant metabolites were selected with a variable importance in projection (VIP) > 1.0 from partial least squares-discriminant analysis (PLS-DA) and a *p*-value < 0.05 from Student’s *t*-test. Integration of transcriptomic and metabolomic data was carried out by mapping differentially expressed genes and significantly altered metabolites to KEGG pathways. Enrichment analysis of pathways was performed using cluster Profiler (v4.6.2) with a significance threshold of adjusted *p*-value < 0.05. Statistical analysis of nutritional composition and growth performance data was conducted using SPSS (v26.0). Data are presented as the mean ± standard deviation (SD). Differences between groups were evaluated by Student’s *t*-test or one-way analysis of variance (ANOVA) followed by Tukey’s post hoc test, with *p* < 0.05 considered statistically significant. All statistical data analyses in this study were performed using Excel 2016 and SAS 9.0 software.

## 3. Results

### 3.1. Key Parameters of Growth, Biomass Conversion and Development in Black Soldier Fly Larvae

Table 3 summarizes the growth performance of black soldier fly larvae from Groups A and C over a 3-day period. The initial body weights at 5 days of age were comparable between the two groups, with Group C (0.0185 ± 0.0061 g) showing a slightly higher mean value than Group A (0.0175 ± 0.0067 g). After 3 days of feeding, the body weights at 8 days of age remained very similar, recorded at 0.0496 ± 0.0158 g for Group A and 0.0491 ± 0.0147 g for Group C. In terms of weight gain, Group A exhibited a marginally higher total weight gain over the 3 days (0.0321 ± 0.0091 g) compared to Group C (0.0306 ± 0.0046 g). Consequently, the average daily weight gain for Group A (0.0106 ± 0.0058 g/day) was also slightly greater than that for Group C (0.0102 ± 0.0058 g/day). Both experimental groups demonstrated satisfactory growth and weight gain during the experimental period. However, Group A, acclimated to Kunming food waste, showed a non-significant trend toward superior performance in both total weight gain and average daily gain compared to Group C, which was acclimated to Qujing food waste.

Body Composition Analysis of Black Soldier Fly Larvae (Table 4). The data indicate that the body composition of the larvae underwent significant changes during development from 5 to 8 days of age, and differences were also observed between groups of different geographical origins (Group A vs. Group C). As the larvae grew, their body composition exhibited systematic changes, with moisture content decreasing in all larvae as age increased. Notably, the crude protein content of Group A larvae increased significantly from 62.73% at 5 days to 66.34% at 8 days, whereas the crude protein content of Group C larvae decreased from 61.59% to 58.10%. In terms of crude fat content, both groups showed an increasing trend with age, but Group C (increasing from 10.67% to 21.62%) demonstrated a significantly higher rate of fat accumulation and final fat content compared to Group A (increasing from 11.37% to 16.49%).In terms of intergroup differences, subtle variations in body composition were already evident at 5 days of age. By 8 days of age, the differences between the two groups became more pronounced: larvae acclimated to Qujing food waste (Group C) tended to accumulate more fat (21.62% vs. 16.49%), while larvae acclimated to Kunming food waste (Group A) accumulated a higher proportion of protein (66.34% vs. 58.10%). Corresponding changes in crude fiber and crude ash content were also observed across different groups and ages.

During the 3-day feeding period, both Group A and Group C larvae consumed 135 kg of food waste (Table 5). Initial weight measurements showed that the total weight of Group A larvae at 5 days was 17.65 ± 6.76 kg, while Group C larvae at 5 days weighed 18.54 ± 6.20 kg. After 3 days of feeding, the total weight of Group A larvae at 8 days increased to 49.67 ± 15.84 kg, and Group C larvae reached 49.16 ± 15.89 kg. Based on this data, Group A larvae gained 32.02 kg over the 3-day period, while Group C gained 30.62 kg. In terms of biomass conversion efficiency, Group A achieved a conversion rate of 23.72%, compared to 22.68% for Group C (Table 6). These results indicate that, with the same amount of food consumption, Group A larvae demonstrated slightly higher growth performance and biomass conversion efficiency.

### 3.2. Analysis of RNA-Seq Gene Results

To investigate the effects of different regional food waste sources on black soldier fly larvae, this transcriptome sequencing study generated approximately 656,482,484 high-quality reads in total, with an average of about 54,706,473 reads per sample. The Q30 score for each sample exceeded 94%, and the alignment efficiency to the reference genome was above 80% across all samples. Approximately 30,000 genes were found to be expressed in at least one sample.

### 3.3. Differential Gene Expression Analysis

Gene expression differed between the two groups. Principal component analysis (PCA) revealed clear separation between groups A (Kunming) and C (Qujing), with biological replicates within each group clustering consistently (Figure 2A). Differential expression analysis under stringent criteria (|log2FoldChange| > 4, *p*-value < 0.01) identified 87 genes significantly upregulated in group A and 73 genes significantly upregulated in group C. Specifically, group A showed marked upregulation of genes encoding detoxification enzymes including cytochrome P450s (e.g., CYP6A1, CYP6g2) and nutrient metabolism proteins such as sterol carrier protein 2 and hexamerin-1.1. Conversely, group C exhibited pronounced upregulation of genes involved in immune response pathways, including multiple lysozyme isoforms and serine proteases (Figure 2B,C). A substantial number of differentially expressed features were identified between groups A and C, with abundant red (upregulated) and blue (downregulated) points concentrated in highly significant regions, indicating extensive and consistent molecular divergence between the two groups. These differences are potentially related to critical pathways including detoxification and nutrient metabolism (Figure 2C).

### 3.4. GO and KEGG Functional Analysis

GO enrichment analysis of differentially expressed genes between groups A and C identified significant enrichment in biological processes including transcription, RNA biosynthesis, vesicle transport, and autophagy (Figure 3A). Molecular function analysis showed predominant enrichment in RNA-binding and enzymatic catalytic activities (Figure 3B). KEGG enrichment analysis of differentially expressed genes revealed that black soldier fly larvae responding to food waste significantly upregulated pathways involved in xenobiotic detoxification, including drug metabolism-cytochrome P450 and glutathione metabolism, while also modulating various nutrient metabolic pathways such as starch and sucrose metabolism and glycerolipid metabolism. Furthermore, the significant enrichment of biosynthesis pathways for secondary metabolites and insect hormones suggests potential molecular responses related to material conversion and growth regulation (Figure 3C). Additionally, fundamental metabolic pathways including carbohydrate metabolism, lipid metabolism, and hormone biosynthesis were differentially enriched between the groups. Analysis of transcription factor families indicated that zf-C2H2 zinc finger proteins represented the most abundant category identified in both groups, followed by bZIP and Homeobox domain-containing factors (Figure 3D).

### 3.5. Metabolome Analysis

Metabolite analysis revealed that carboxylic acids and their derivatives constituted the largest proportion of compounds (19.94%), playing a dominant role in energy metabolism. Unidentified metabolites accounted for 13.86%, which may imply unique adaptive mechanisms and also serve as potential targets for elucidating region-specific acclimation processes. Intermediate proportions of organic oxygen compounds, fatty acyl groups, and other components supported essential metabolic and detoxification functions, while Diazines were present in minimal amounts (Figure 4A). PLS-DA results demonstrated clear separation between samples from groups A and C, confirming that acclimation to regional food waste sources has led to stable and significant molecular differentiation between the two groups. This separation not only serves as a basis for discrimination but also guides further investigation into key differential metabolites. Although some individual variation was observed within groups, the overall clustering trend remained distinct (Figure 4B). Chord diagram visualization of superclass distribution and associations showed that undefined categories largely consist of unannotated metabolites, which may include unique products induced by processing of regional food waste. The prominence of lipids and organic acids suggests active energy metabolism and fundamental biosynthetic processes, while minor components such as alkaloids and benzenoids may be involved in secondary metabolism and detoxification adaptation. Inter-category associations reflect cooperative interactions within the metabolic network (Figure 4C).

In response to distinct regional food waste sources, black soldier flies developed specific metabolic pathway adaptation strategies: Group A enhanced pathways such as branched-chain amino acid degradation and phenylalanine metabolism, enabling efficient utilization of protein and aromatic nutrients present in Kunming food waste. In contrast, Group C activated pathways including sphingolipid signaling and specific amino acid metabolism to cope with stress stimuli or nitrogen metabolism demands associated with Qujing food waste. Most differentially regulated pathways contained between 5 to 10 metabolites, suggesting that coordinated changes in core metabolites drive the observed inter-group differences (Figure 5A). Regional dietary differences drove significant adjustments in amino acid metabolism pathways (e.g., leucine and glutamine) in both Groups A and C, reflecting divergent strategies in nitrogen utilization and energy allocation. The elevated expression of indole derivatives in Group C may result from higher stressor content in Qujing food waste, promoting its synthesis to activate defense-related pathways (Figure 5B). Known classes such as lipids, benzenoids, and organic acids dominated core metabolic processes. Close associations between alkaloids and benzenoids indicate synergistic roles in defense and detoxification, while interactions between unknown metabolites and lipids/organic acids may reflect their involvement in energy reallocation to optimize utilization of food waste nutrients.

## 4. Discussion

This study employs integrated transcriptomic and metabolomic analyses to elucidate the molecular mechanisms underlying the adaptive differentiation of black soldier fly larvae to region-specific food waste. Partial least squares-discriminant analysis (PLS-DA) revealed clear separation between the two groups, confirming that long-term adaptation to distinct regional food waste sources has led to stable molecular differentiation. While growth rates were comparable, Kunming-adapted larvae (Group A) exhibited higher biomass conversion efficiency (23.72% vs. 22.68%) with a propensity for protein accumulation (crude protein 66.34%), whereas Qujing-adapted larvae (Group C) demonstrated enhanced lipid storage (crude fat 21.62%). These divergent nutrient allocation strategies correspond to the distinct characteristics of regional food waste: Kunming waste being rich in proteins and lipids, while Qujing waste is predominantly plant-based.

Transcriptomic analysis revealed differential gene expression between Group A and Group C, with the differentially expressed genes primarily enriched in pathways related to xenobiotic metabolism, nutrient sensing, and transcriptional regulation. This result is associated with the adaptation of larvae to the characteristics of food waste from different regions [14,15]. The upregulation of CYP6 gene transcription was linked to the metabolism of xenobiotic compounds in the diet, supporting the conserved role of cytochrome P450 enzymes, particularly the CYP6 subfamily, in the metabolism of aromatic compounds in insects [16,17]. Additionally, zf-C2H2 zinc finger transcription factors were identified as upstream regulators of CYP450 expression, suggesting an adaptive mechanism extending from effector molecules to regulatory networks [18,19,20], This regulatory pattern is consistent with the involvement of the C2H2 family in environmental responses in some insect species [21,22,23].

In terms of nutrient sensing, the higher expression of TOR-S6K pathway genes in Group A may promote the degradation of branched-chain amino acids, aligning with the high-protein dietary characteristics, and reflecting host-autonomous metabolic regulation during the adaptation of *Hermetia illucens* [24,25]. Metabolomic analysis indicated that the two groups of larvae adopted distinct metabolic strategies. In Group A, enhanced degradation of branched-chain amino acids and increased phenylalanine hydroxylase activity contributed to the efficient utilization of proteins and aromatic nutrients in the Kunming food waste [26,27]. In contrast, Group C activated the sphingolipid signaling pathway and accumulated indole derivatives. Studies have shown that indoles may participate in antioxidant responses by activating the aryl hydrocarbon receptor (AhR) pathway, providing a new perspective for understanding the adaptive mechanisms of insects under dietary stress [28,29]. This differential regulation of amino acid metabolism represents an adaptive strategy tailored to local dietary conditions, ensuring metabolic flexibility and promoting survival under diverse nutritional environments [30,31]. In the metabolomic data, 13.86% of the metabolites remained unannotated, but these compounds exhibited high connectivity in the metabolic network, significantly exceeding random network levels (*p* < 0.01), suggesting their potential role as “bridge metabolites” in metabolic regulation. These unknown metabolites may be associated with *Hermetia illucens*-specific metabolic processes, such as antimicrobial peptide synthesis or xenobiotic compound metabolism, and are closely linked to lipid and organic acid metabolism, indicating their potential roles in energy redistribution and efficient nutrient utilization [32,33].

## 5. Conclusions

This study employed integrated transcriptomic and metabolomic analyses to elucidate the adaptation mechanisms of black soldier flies to regional food waste, providing a theoretical basis for optimizing their bioconversion efficiency. The findings suggest that zf-C2H2 transcription factors may regulate detoxification genes, the indole-AhR pathway plays a role in stress response, and the Kunming population (Group A) exhibited enhanced branched-chain amino acid metabolism. These results indicate that breeding strains with enhanced activity in specific metabolic pathways or optimizing feed formulations to activate relevant metabolic mechanisms could improve protein accumulation efficiency in black soldier fly larvae. However, this study has several limitations: the geographical sampling was confined to the Yunnan–Guizhou Plateau region, functional validation of key genes remains incomplete, and gut microbiome data were not integrated. Future research could expand sampling to coastal and industrial areas, incorporate whole-genome resequencing technologies, and further investigate host–microbiome metabolic interactions to provide a more comprehensive scientific foundation for the resource utilization of black soldier flies.

## Figures and Tables

**Figure 1 biology-14-01584-f001:**
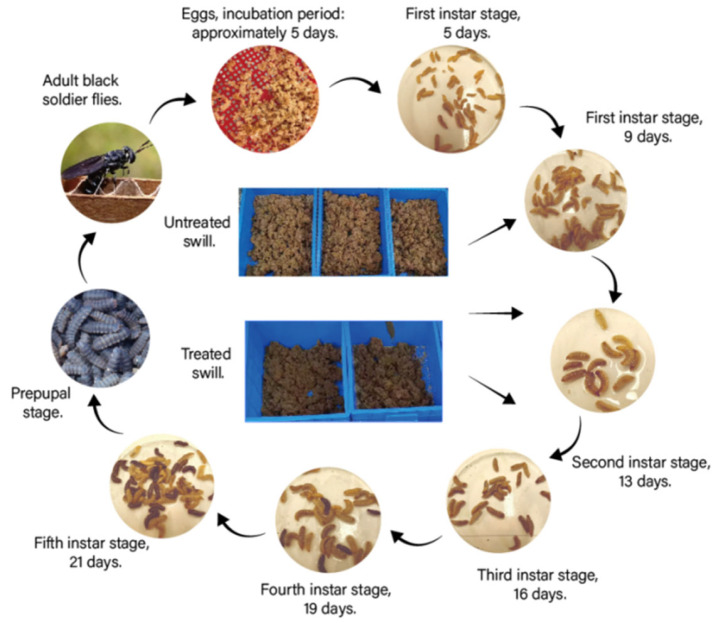
Black soldier fly larvae at different growth stages after processing food waste.

**Figure 2 biology-14-01584-f002:**
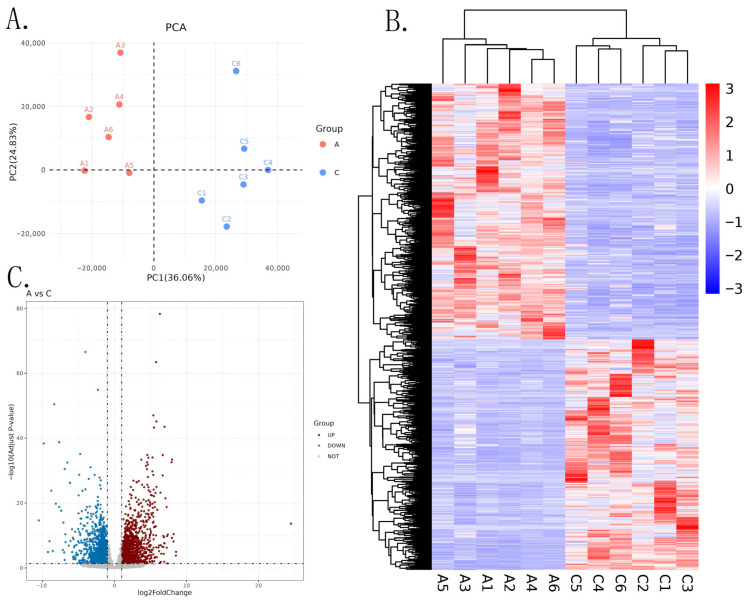
(**A**) Principal Component Analysis (PCA) Scatter Plot. In the legend on the right, red dots represent Group A, and blue dots represent Group C. The horizontal axis (PC1) represents the first principal component, explaining 36.06% of the data variation. It is the primary factor contributing to intergroup differences, with larger values indicating more pronounced sample characteristics along this principal component dimension. The vertical axis (PC2) represents the second principal component, explaining 24.83% of the data variation, serving as the secondary factor for differences. (**B**) Heatmap. In the color bar on the right, red represents high expression, blue represents low expression, and white indicates values close to 0, reflecting the relative trends of the data. Horizontally: The sample labels of Group A and Group C are annotated at the bottom, representing biological samples under different treatments. Vertically: The black dendrogram on the left clusters “features” based on similarity in expression patterns, with each row representing the expression changes of one feature. (**C**) Volcano Plot of differential expression analysis, used to display molecular-level differences between Group A and Group C of black soldier flies. The horizontal axis represents the fold change of Group A relative to Group C. Positive values indicate that the expression/abundance of the feature in Group A is higher than in Group C, meaning Group A is twice that of Group C; negative values indicate lower expression/abundance in Group A. The vertical axis represents the significance level of the statistical test. Larger values indicate more significant differences. Red (UP): represents features significantly upregulated in Group A. Blue (DOWN): represents features significantly downregulated in Group A. Gray (NOT): features that do not meet the criteria for significant differences, indicating no significant intergroup differences.

**Figure 3 biology-14-01584-f003:**
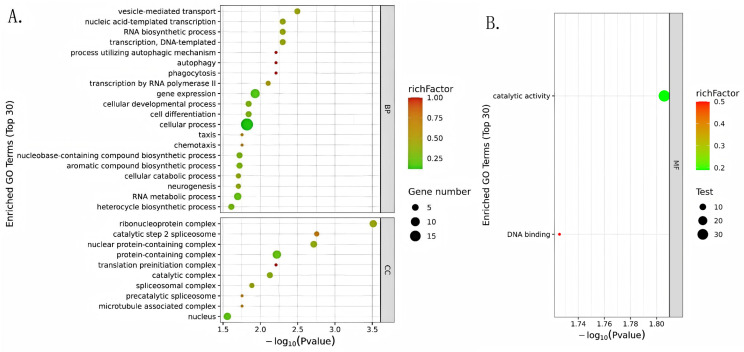
(**A**) Visualization of Gene Ontology (GO) Enrichment Analysis. Horizontal axis: negative logarithm transformation of the *p*-value; larger values indicate higher enrichment significance. Vertical axis (Enriched GO Terms (Top 30)): the top 30 significantly enriched GO terms, ranked by enrichment significance. Each dot represents one GO term. Dot attributes: Color: determined by the richFactor, with red → yellow → green → blue indicating a gradual decrease in richFactor. The richFactor = the number of differential genes belonging to that GO term; a larger value indicates a higher enrichment degree of the GO term among the differential genes. Size: determined by the Gene number; larger dots indicate that the GO term contains more differential genes. (**B**) Visualization of the “Molecular Function (MF)” dimension in GO enrichment analysis, used to display the enrichment of differential genes at the molecular function level. Axes: The horizontal axis represents the negative logarithm of the *p*-value from the enrichment analysis; larger values indicate higher statistical significance of enrichment. The vertical axis (Enriched GO Terms (Top 30)): displays significantly enriched GO terms, here showing only two main molecular function entries, “catalytic activity” and “DNA binding”, ranked by enrichment significance. Legend: Color: determined by the richFactor, where richFactor = (number of differential genes belonging to the GO term)/(number of background genes belonging to the GO term); a larger value indicates a higher enrichment degree of this molecular function among the differential genes. Size: determined by the Test; larger dots indicate that the molecular function involves more differential genes. Right annotation (MF): specifies that the figure corresponds to the “Molecular Function” dimension of GO classification, which, together with BP (Biological Process) and CC (Cellular Component), constitutes a complete GO analysis. (**C**) KEGG Pathway Enrichment Analysis diagram. Horizontal axis: negative logarithm transformation of the *p*-value; larger values indicate higher significance of pathway enrichment. Vertical axis (KEGG Pathways (Top 20)): the top 20 significantly enriched KEGG pathways, ranked by enrichment significance. Each dot represents one KEGG pathway. Dot attributes: Color: determined by the Rich factor, with red → yellow → green → blue indicating a gradual decrease in Rich factor. Rich factor = the number of differential genes belonging to the pathway; a larger value indicates a higher enrichment degree of the pathway among the differential genes. Size: determined by the Gene number; larger dots indicate that the pathway contains more differential genes. (**D**) Distribution pie chart of Transcription Factor Domains, showing the proportion of different types of transcription factor domains in the sample. Domain types: Each segment represents one characteristic domain of a transcription factor. Proportion and significance: The area proportion of the pie chart reflects the relative abundance of that domain type among all transcription factors.

**Figure 4 biology-14-01584-f004:**
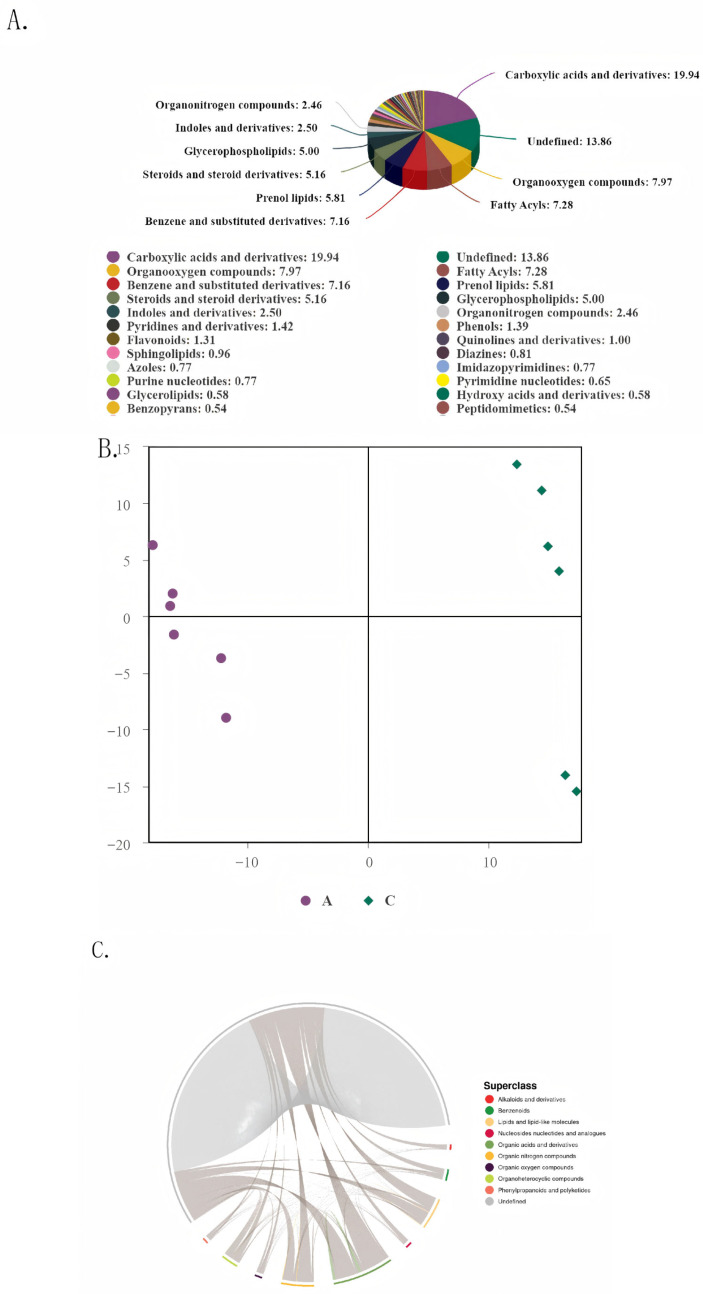
(**A**) Metabolite Category Distribution Pie Chart, showing the relative proportions of different metabolite categories. (**B**) Partial Least Squares-Discriminant Analysis (PLS-DA) Visualization Plot. Group identifiers: In the legend below, purple dots represent Group A, and green diamonds represent Group C. Distribution characteristics: Samples from Group A are concentrated in the negative direction of the horizontal axis, while samples from Group C are concentrated in the positive direction of the horizontal axis. (**C**) Chord Diagram showing the distribution of metabolite classifications by “Superclass”. Core elements: Chords (gray lines) connect different metabolite superclasses, representing the relationships between categories; the thickness of the lines reflects the “association strength”. Segments: Each segment represents one superclass, and its area reflects the proportion of that category among all metabolites. The legend on the right annotates the 11 metabolite superclasses: Phenylpropanoids (red), primarily involved in plant defense and signaling; Benzenoids (green), which form the basic structures of aromatic compounds; Lipids and lipid-like molecules (yellow), associated with energy storage and membrane structures; Undefined (gray), representing the largest proportion, denotes metabolites with undefined structures.

**Figure 5 biology-14-01584-f005:**
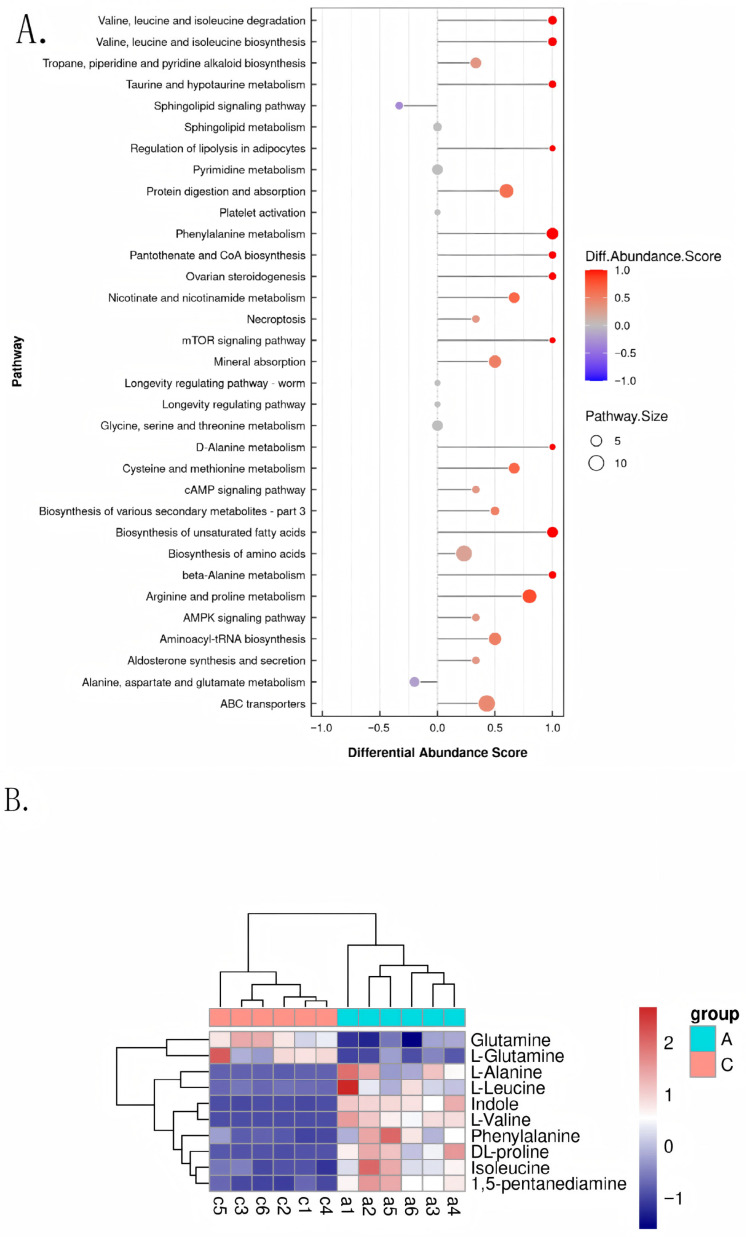
(**A**) Differential Abundance Analysis of Metabolic Pathways. Horizontal axis (Differential Abundance Score): pathway differential abundance score. Positive values indicate higher abundance of the pathway in Group A compared to Group C (red), while negative values indicate higher abundance in Group C (blue). Larger absolute values indicate more significant differences. Vertical axis (Pathway): names of metabolic pathways ranked by differential abundance, displaying the most differentially abundant key pathways. Color: determined by the “Diff.Abundance.Score” color bar on the right; red indicates higher abundance in Group A, blue indicates higher abundance in Group C, and the color intensity reflects the degree of difference. Size: determined by “Pathway.Size”; larger circles indicate that the pathway contains more metabolites. Position: the horizontal position of each point corresponds to the differential abundance score, and the vertical position corresponds to the pathway name. Points closer to the ends indicate more significant pathway differences. (**B**) Clustered Heatmap of Differential Metabolite Expression, incorporating hierarchical clustering. Displays expression differences and clustering patterns of metabolites such as amino acids and indoles between Group A and Group C of black soldier flies. Horizontal axis: sample labels of Group A and Group C are annotated at the bottom. Vertical axis (Metabolites): lists the differential metabolites, which are the core indicators of the analysis. Group identifiers: the “group” legend on the right, where cyan represents Group A and pink represents Group C; the “color bar” in red-white-blue corresponds to high to low metabolite expression levels.

**Table 1 biology-14-01584-t001:** Comparison of Basic Characteristics of Food Waste.

Characteristic	Kunming Panlong District Food Waste	Qujing Qilin District Food Waste
Primary Source	Large Restaurants	Households and Small-to-Medium Restaurants
Moisture Content (wet basis)	75–85%	70–80%
Organic Matter Content (dry basis)	80–93%	75–85%
Oil & Fat Content (wet basis)	3–5%	2–4%
Salt Content (wet basis)	1.5–2.5%	1.0–1.5%

**Table 2 biology-14-01584-t002:** Comparison of Organic Matter Composition in Food Waste (dry basis).

Organic Component	Kunming Panlong District Food Waste	Qujing Qilin District Food Waste
Carbohydrates (Starch, Cellulose)	40–50%	60–70%
Protein	15–20%	10–15%
Others (Primarily starchy materials)	30–45%	20–25%

**Table 3 biology-14-01584-t003:** Growth Rate of Black Soldier Fly Larvae.

Group	Body Weight at 5 Days (g)	Body Weight at 8 Days (g)	Weight Gain over 3 Days (g)	Average Daily Weight Gain (g)
Group A	0.0175 ± 0.0067	0.0496 ± 0.0158	0.0321 ± 0.0091	0.0106 ± 0.0058
Group C	0.0185 ± 0.0061	0.0491 ± 0.0147	0.0306 ± 0.0046	0.0102 ± 0.0058

**Table 4 biology-14-01584-t004:** Nutrient Composition of Black Soldier Fly Larvae.

Nutrient	Group A (5d)	Group C (5d)	Group A (8d)	Group C (8d)
Moisture	78.49 ± 0.71	78.54 ± 0.66	73.32 ± 0.61	69.93 ± 0.76
Crude Protein	62.73 ± 0.21	61.59 ± 0.33	66.34 ± 0.68	58.10 ± 0.75
Crude Fat	11.37 ± 0.51	10.67 ± 0.77	16.49 ± 0.55	21.62 ± 0.63
Crude Fiber	8.74 ± 0.36	6.80 ± 0.41	9.75 ± 0.50	10.97 ± 0.48
Crude Ash	9.95 ± 0.62	9.54 ± 0.57	7.12 ± 0.62	7.65 ± 0.72

Note: Moisture content is reported on a fresh weight basis; all other nutrient values are expressed on a dry matter basis (%).

**Table 5 biology-14-01584-t005:** Food waste consumption and growth performance of black soldier fly larvae over a 3-day period.

3-Day Food Waste Consumption (kg)	Larval Weight at 5 Days (kg)		Larval Weight at 8 Days (kg)	
135	Group A	Group C	Group A	Group C
	17.65 ± 6.76	18.54 ± 6.20	49.67 ± 15.84	49.16 ± 15.89

**Table 6 biology-14-01584-t006:** Biomass Conversion Efficiency.

Conversion Rate Calculation Formula	Group A	Group C
Efficiency Conversion Rate = [(F − E)/D] × 100%	23.78%	22.68%

Note: E: Larval weight at 5 days; F: Larval weight at 8 days; D: 3-day food waste consumption.

## Data Availability

The datasets presented in this study can be found in online repositories. The names of the repository and accession number(s) can be found in the NCBI SRA database with accession numbers PRJNA1159383 (https://dataview.ncbi.nlm.nih.gov/object/PRJNA%201159383?%20reviewer%20=%20r686p3tg9m4iddtoak0bb2og8g (accessed on 24 September 2025).

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
